# Forecasting Principles from Experience with Forecasting Competitions

Jennifer L. Castle [1], Jurgen A. Doornik [2,*] and David F. Hendry [2]

1 Magdalen College and Institute for New Economic Thinking at the Oxford Martin School, University of Oxford, High Street, Oxford OX1 4AU, UK; jennifer.castle@magd.ox.ac.uk
2 Institute for New Economic Thinking at the Oxford Martin School, and Climate Econometrics, Nuffield College, University of Oxford, New Road, Oxford OX1 1NF, UK; david.hendry@nuffield.ox.ac.uk
* Correspondence: jurgen.doornik@nuffield.ox.ac.uk

**Abstract:** Economic forecasting is difficult, largely because of the many sources of nonstationarity influencing observational time series. Forecasting competitions aim to improve the practice of economic forecasting by providing very large data sets on which the efficacy of forecasting methods can be evaluated. We consider the general principles that seem to be the foundation for successful forecasting, and show how these are relevant for methods that did well in the M4 competition. We establish some general properties of the M4 data set, which we use to improve the basic benchmark methods, as well as the Card method that we created for our submission to that competition. A data generation process is proposed that captures the salient features of the annual data in M4.

**Keywords:** automatic forecasting; calibration; prediction intervals; regression; forecasting competitions; seasonality; software; time series; nonstationarity

## 1. Introduction

Economic forecasting is challenging—and always has been. No clear consensus as to the 'best' or even 'good' approaches has arisen in the literature. Sophisticated methods often fail to beat a simple autoregressive model. Even when a small advantage is shown, this may fail to survive in slightly different settings or time periods. What does hold is that all theoretical results that assume stationarity are irrelevant. Instead, small shocks occur regularly and their effects can cumulate, and large shocks and structural breaks happen intermittently. Causes of such breaks can be financial crises, trade wars, conflicts, policy changes, and pandemics etc. Consequently, nonstationarities can arise from both unit roots and structural breaks in mean or variance.

The sequence of forecasting 'competitions' commencing with [1] analysing 111 time series, through [2–4] with M3, then M4 (see [5], held by the M Open Forecasting Center at the University of Nicosia in Cyprus) created realistic, immutable, and shared data sets that could be used as testbeds for forecasting methods. Ref. [6] provides a more general history and [7] consider their role in improving forecasting practice and research. However, interpretations of the findings from the competitions have been ambiguous as exemplified by the different conclusions drawn by [4,8,9]. There are also limitations to the data: variables are anonymized, and have unknown and differing sample periods. This prevents the use of subject matter expertise or any judgemental adjustment irrespective of skill or knowledge (as analyzed by e.g., [10]). It also creates problems for methods that try to link variables for forecasting: the misaligned sample periods may cause some variables to be ahead in time of others. Unfortunately, this also rules out any multiple variable or factor approaches. Nevertheless, such forecasting competitions have made valuable contributions to improving the quality of economic forecasting methods, as well as increasing our understanding of methods and techniques. A useful addition to M4 was the request for interval forecasts: good expressions of the uncertainty of a forecast are just

as important as the forecasts themselves, a feature emphasized by the 'fan charts' used by the Bank of England in their *Inflation Reports*: see [11]. Probabilities of rain are now a routine aspect of weather forecasts, and forecast uncertainty should be quantified in other settings as well.

By combining the insights from our research, the literature, M4 results, and forecasting the COVID-19 pandemic, there seem to emerge a few relatively general 'principles' for economic forecasting:

(I)       dampen trends/growth rates;
(II)      average across forecasts from 'non-poisonous' methods;
(III)     include forecasts from robust devices in that average;
(IV)     select variables in forecasting models at a loose significance;
(V)      update estimates as data arrive, especially after forecast failure;
(VI)     'shrink' estimates of autoregressive parameters in small samples;
(VII)    adapt choice of predictors to data frequency;
(VIII)   address 'special features' like seasonality.

The authors made a submission to the M4 competition that did well in many aspects. Here we aim to interpret such results, and those of some methods that did well in M3 and M4, in light of these principles. As part of this, we first summarize the properties of the M4 data set, which leads to improved benchmark forecast methods.

Our forecast method, *Card*, is formally described in [12]. The procedure is based on simple autoregressive models, augmented with a damped trend (I) (see e.g., [13], followed by many applications) and some robustification through differencing (III), as our earlier research has emphasised the key role of location shifts in forecast failure (see [14,15]). Forecast combination (II) is also used as it is found to be risk reducing and can even outperform the best of the methods averaged over (dating from [16], and widely used). 'Wild' forecasts from small data samples are monitored (VI), and seasonality is handled (VIII). As only univariate methods seemed feasible, (IV) was not relevant (see [17], for an analysis), although some entrants used the data in other series which may lead to infeasible methods. Also note that [18] show that additional explanatory variables in models need not improve forecasts even when their future values are known. (V) could not be implemented given the structure of the competitions (see e.g., [19], for its advantages following a break in the included variables), although we did graph recursive forecasts. Finally, we know (VII) matters, as different entries performed best at different data frequencies. Below we show that some further improvement can be made by paying more attention to our principles. Furthermore, an improved formulation of the forecast standard errors is provided for our method.

M4 is the fourth generation of M forecast competitions, created by Spyros Makridakis and the M4 team. M4 provides a database of 100,000 series requiring out-of-sample forecasts. This large size makes it a computational challenge too. Efficient production of forecasts is useful, even more so when studying subsample properties. The previous competition, M3, consisted of "only" 3003 variables. The best performing method in the M3 competition is the so-called *Theta* method, see [4] and Section 2.2 below, so many papers take that as the benchmark to beat. Ref. [20] demonstrated the non-invariance of evaluation metrics to linear transformations that leave forecasting models invariant, but the choice was set by the organisers and is discussed in Section 2.4.

Proper handling of seasonality is expected to be an important aspect of forecasting. This implies that the exercise has some similarities to the X12-ARIMA (autoregressive-integrated moving average) programme of the US Census Bureau, see [21]. The X12 approach involves estimating a seasonal ARIMA model to extend the series with forecasts, followed by smoothing using a sequence of moving averages in the seasonal and deseasonalized directions. Estimated moving averages are sensitive to outliers and structural breaks, and procedures need to make allowance for this.

The remainder of this paper is as follows. We discuss the structure of M4 and its benchmark methods in Section 2, and we describe M4 data properties in Section 3. Next, we

adapt the *Theta* method, and introduce a simple but effective variant in Section 4. We also consider the expected outcomes of the accuracy measures and study the performance of the new benchmark methods. In Section 5 we discuss how the heterogeneity and independence of time-series samples affects outcomes and propose a simulation experiment that captures the salient features of yearly M4. In Section 6 we consider improvements to the *Card* method, leading to *Cardt*. This plays a prominent role in our COVID-19 forecasts, see Section 8. Section 7 evaluates *Cardt* with the M3 and M4 data. Finally, Section 9 concludes. Derivations and supplementary results are presented in appendices.

## 2. M4 Competition

We compare the design of the M4 competition to that of M3 in Section 2.1. The objective of M4 is to mimimize two error measures, one for the point forecasts, and another for the interval forecasts, Section 2.4. The former, denoted overall weighted average (OWA), standardizes using one of the benchmark methods. Hence, these are described in Section 2.2 for annual data, with adjustments for seasonality in Section 2.3.

### 2.1. Overview

In many respects, M4 is a direct successor to the M3 competition. A range of annual, quarterly, and monthly time series is provided with the aim to make out-of-sample forecasts for each of these series up to a specified horizon. These forecasts are evaluated against a sample that has been held back by the organizers. The time series are anonymized: name and sample dates are unknown, and the series are rescaled. The variables are categorized as demographic, finance, industry, macro, micro, and other. Most forecast applications ignore these categories. Evaluation is on a selection of performance metrics, such as variants of the mean absolute percentage error (MAPE). The M3 data set has become widely used as a benchmark for new forecast methods.

There are some differences between M3 and M4 as well. M4 is much larger at 100,000 time series, whereas M3 has 3003. M4 introduces weekly, daily, and hourly data; M3 has a small set with "other" frequency. M4 includes a possibility to report interval forecasts. By encouraging participants to publish their methods on Github, M4 achieves a greater level of replicability.

M4 was run as a proper competition, with monetary prizes for those methods scoring best on a single criterion, separately for the point forecasts and intervals. M3, on the other hand, evaluated the forecasts on five criteria, but did not produce an overall ranking. M3 also included an opportunity to receive interim feedback on the forecast performance, as reported by [22].

M4 addresses several of the shortcomings that were raised in the special issue on M3 of the *International Journal of Forecasting* (2001, 17): addition of weekly data and interval forecasts, more seasonal data, and potentially a more representative data set through the vast number of series. The M3 report did not include a statistical test to see whether forecasts are significantly different. This was added later by [23], and included in the M4 report, although not split by frequency.

Other shortcomings remain. Different performance metrics will result in different rankings. The series are all positive, and forecasting growth rates may also completely change the ranking of forecasting procedures. Finally, the design is inherently univariate. No sample information is available, so some series may contain future information on other series. As a consequence, methods that use other series may inadvertently be infeasible: in practice we never know future outcomes, except for deterministic or predetermined variables.

Some basic aspects of the M4 data are given in Table 1, including values of the forecast horizon *H* that are specific to each frequency. The yearly, monthly, and quarterly series together constitute 95% of the sample, so will dominate the overall results. The frequency is given through the labels "hourly", "weekly" etc. but not otherwise provided. Hence, daily data could be for five weekdays or a full seven day week. The evaluation frequency

$m$ is used in the performance measures (Section 2.4). It is left to the participants to choose the frequency $S$ (or dual frequencies $S$ and $S_2$) for their methods.

Table 1 records the lengths of shortest and longest series under $T_{\min}$ and $T_{\max}$. This is for the competition (or training) version of the data, so excluding the $H$ observations that were held back until after the competition. The sample sizes of annual data range from 13 to 835, but it is unlikely that there is a benefit from using information of more than 800 'years' ago.

**Table 1.** Basic properties of the M4 data set.

| | Dimension | | Evaluation | Sample Size | | Forecasts |
|---|---|---|---|---|---|---|
| | **# Series** | **%** | **$m$** | **$T_{\min}$** | **$T_{\max}$** | **$H$** |
| Yearly | 23,000 | 23.0% | 1 | 13 | 835 | 6 |
| Quarterly | 24,000 | 24.0% | 4 | 16 | 866 | 8 |
| Monthly | 48,000 | 48.0% | 12 | 42 | 2794 | 18 |
| Weekly | 359 | 0.4% | 1 | 80 | 2597 | 13 |
| Daily | 4227 | 4.2% | 1 | 93 | 9919 | 14 |
| Hourly | 414 | 0.4% | 24 | 700 | 960 | 48 |

### 2.2. M4 Benchmark Forecasting Methods

The organizers of M4 used ten forecast methods as benchmark methods: these would be evaluated and included in the final results by the organizers. These benchmark methods comprised three random-walk type extrapolation methods, five based on exponential smoothing, and two machine learning methods. We review random walk, exponential smoothing, and *Theta* forecasts, but ignore the basic machine learning and neural network approaches. Ref. [24] show the inferior forecasting performance of some machine learning and artificial intelligence methods on monthly M3 data.

The random walk forecasts of annual and nonseasonal data are a simple extrapolation of the last observation. This is called *Naive2* forecasts in M4 (and the same as *Naive1* in the absence of seasonality). Denoting the target series as $y_1, \ldots, y_T$, the objective is to forecast $y_{T+1}, \ldots, y_{T+H}$, in this case:

$$\widehat{y}_{T+h} = y_T, h = 1, \ldots, H.$$

Exponential smoothing (ES) methods are implemented as single source of innovation models, see [25,26]. They can be formulated using an additive or multiplicative (or mixed) representation. The additive exponential smoothing (AES) model has the following recursive structure:

$$\mu_t = l_{t-1} + b_{t-1},$$
$$\epsilon_t = y_t - \mu_t,$$
$$l_t = l_{t-1} + \alpha\epsilon_t,$$
$$b_t = b_{t-1} + \delta\epsilon_t,$$

where $y_t$ is the observed time series, $\epsilon_t$ the one-step prediction error, $l_t$ the level, and $b_t$ the slope. Given initial conditions $l_0$ and $b_0$, the coefficients $\alpha$ and $\delta$ can be estimated by maximum likelihood. An alternative approach is to add the initial conditions as additional parameters for estimation. Forecasting simply continues the recursion with $\epsilon_t = 0$, keeping the parameters fixed.

AES includes the following forecasting methods, among others:

| | | |
|---|---|---|
| *SES* | Simple exponential smoothing | $\delta = b_0 = 0$, |
| *HES* | Holt's exponential smoothing, | |
| *Theta2* | Theta(2) method | $\delta = 0, b_0 = \widehat{\tau}/2$ defined in (1). |

The *SES* and *HES* models with infinite startup are ARIMA$(0, 1, 1)$ and ARIMA$(0, 2, 2)$ models respectively, see [25] (Ch.11). A dampened trend model adjusts the slope equation

to $b_t = \psi b_{t-1} + \delta \epsilon_t$ where $\psi < 1$. Holt–Winters exponential smoothing adds a seasonal equation to the system (see [27,28]).

The *Theta* method of [29] first estimates a linear trend model

$$y_t = \mu + \tau(t-1) + u_t, \ t = 1, \ldots, T, \tag{1}$$

by OLS. The *Theta* forecasts are then the sum of the extrapolated trend and forecasts from the model for $y_t(\theta) = y_t - \widehat{\tau}(t-1)/\theta$, with weights $1/\theta$ and one respectively. The suggested model for $y_t(\theta)$ is SES, in which case this method can be implemented within the AES framework, as shown by [30]. *Theta2*, i.e., using $\theta = 2$, had the best sMAPE (see Section 2.4 below) in the M3 competition, see [4]. It is also the best benchmark method by the overall criterion (5), so we ignore the others.

The AES estimates depend on several factors: initial conditions of the recursion, imposition of parameter constraints, and objective function. We have adopted different conventions for the initial conditions, so, in general, will get different results from the R forecast package ([31]). The exception to this is *SES* with $0.001 \leq \alpha \leq 0.9999$ and $l_0$ as an estimated parameter. We also get almost identical results for *Theta2*. We impose $0.001 \leq \alpha \leq 0.9999$, and set $l_0 = y_1 - b_0$, which conditions on the first observation to force $\epsilon_1 = 0$. For the yearly M3 data with $H = 6$ we obtain an sMAPE of 16.72, where [30] (Table 1) report 16.62 (the submission to M3 has sMAPE of 16.97).

### 2.3. Seasonality in the M4 Benchmark Methods

Let $r_i$ denote the terms of the autocorrelation function (ACF) of a time series with $T$ observations. The seasonality decision in the M4 benchmarks for frequency $S > 1$ is based on the squared $S$th autocorrelation of the series $y_t, t = 1, \ldots, T$:

$$R(S) = T \frac{r_S^2}{1 + 2\sum_{j=1}^{S-1} r_i^2} \sim \chi^2(1). \tag{2}$$

If this test of nonseasonality rejects with a $p$-value of 10% or less, the series is seasonally adjusted with an $\text{MA}_{2 \times S}$ filter (or just $\text{MA}_S$ if $S$ is odd). The benchmark is applied to the seasonally adjusted series, and the forecasts are then recolored with the seasonality. The seasonal adjustments use multiplicative adjustment throughout.

Assuming the frequency $S$ is even, the seasonal component is the deviation from a smooth 'trend':

$$s_t = y_t / \text{MA}_{2 \times S}(y_t), t = 1 + S/2, \ldots, T - S/2.$$

Now let $\bar{s}_{.j}$ denote the average for each season $j = 1, \ldots, S$ from the $T - S$ observations $s_t$ (so not all seasons need to have the same number of observations). These seasonal estimates are normalized to the frequency:

$$\widehat{s}_j = \frac{\bar{s}_{.j}}{S} \sum_{k=1}^{S} \bar{s}_{.k}.$$

The seasonally adjusted series is

$$y_t^{\text{sa}} = y_{i,j} / \widehat{s}_j.$$

Finally, the forecasts from the seasonally adjusted series are multiplied by the appropriate seasonal factors. For the *Naive2* method, assuming observation $y_T$ is for period $S$, the forecasts are:

$$y_T^{\text{sa}} \widehat{s}_1, \ y_T^{\text{sa}} \widehat{s}_2, \ \ldots, \ y_T^{\text{sa}} \widehat{s}_S, \ y_T^{\text{sa}} \widehat{s}_1, \ \ldots.$$

*2.4. M4 Forecast Evaluation*

M4 uses two scoring measures, called MASE ([32]) and sMAPE ([33]). For time-series $y_t, t = 1, \ldots, T + H$ with forecasts $\widehat{y}_t$ produced over $T + 1, \ldots, T + H$ and seasonal frequency $m$ (which is allowed to differ from the frequency $S$ for seasonal adjustment):

$$\text{sMAPE} = \frac{100}{H} \sum_{t=T+1}^{T+H} \frac{|y_t - \widehat{y}_t|}{(|y_t| + |\widehat{y}_t|)/2}, \tag{3}$$

$$\text{MASE} = \frac{1}{H} \frac{\sum_{t=T+1}^{T+H} |y_t - \widehat{y}_t|}{\overline{|\Delta_m y|}}. \tag{4}$$

The denominator of MASE is the average of the seasonal difference over the 'estimation period':

$$\overline{|\Delta_m y|} = \frac{1}{T - m} \sum_{t=m+1}^{T} |y_t - y_{t-m}|.$$

MASE is infinite if the series is constant within each season: in that case we set it to zero.

Next, the sMAPE and MASE are each averaged over all forecasts in the evaluation set. To facilitate comparison, these averages can be scaled by the average accuracy of the benchmark *Naive2* forecasts. This leads to the final objective criterion of M4 for method $X$:

$$\text{OWA} = \frac{1}{2} \frac{\overline{\text{sMAPE}(X)}}{\text{sMAPE(Naive2)}} + \frac{1}{2} \frac{\overline{\text{MASE}(X)}}{\text{MASE(Naive2)}} \tag{5}$$

A $100(1 - \alpha)\%$ forecast interval is expressed as $[\widehat{L}_t, \widehat{U}_t]$. Accuracy of each forecast interval for all series and horizons $h$ is assessed on the basis of mean scaled interval score (MSIS):

$$\text{MSIS} = \frac{1}{\overline{\Delta_m y}} \frac{1}{H} \sum_{t=T+1}^{T+H} \left[ \widehat{U}_t - \widehat{L}_t + \frac{2}{\alpha} (\widehat{L}_t - y_t) I(y_t < \widehat{L}_t) + \frac{2}{\alpha} (y_t - \widehat{U}_t) I(y_t > \widehat{U}_t) \right]. \tag{6}$$

M4 uses $\alpha = 0.05$, so that any amount outside the bands is penalized by forty times that amount. Ref. [34] (p. 370) show that the interval score is 'proper,' meaning that it is optimized at the true quantiles.

We can also count the number of outcomes that are outside the given forecast interval. For a 95% pointwise interval, we aim to be outside in about 5% of cases, corresponding to a coverage difference of close to zero.

## 3. M4 Data

The large dimension of the M4 data set makes it impossible to look at all series in detail. But a limited exploration makes it clear that most variables appear to be in levels, and many have strong signs of seasonality. This section focuses on time-series properties that may affect the implementation of forecast procedures: sample size (Section 3.2), using logarithms (Section 3.3), persistence (Section 3.4), and testing seasonality (Section 3.5).

All results in this section are based on the training data, so exclude the sample that was held back to evaluate the submitted forecasts.

*3.1. Properties of Interest*

Understanding the properties of large datasets is a challenge. Ref. [35] propose a solution for M3, which is applied to M4 in [36]. They characterize each series on six aspects, each on a scale of zero to one. This is then summarized by the first two principal components. If we ignore the estimate of the seasonal periodicity, then three aspects are based on an additive decomposition in trend, seasonal, and residual: relative variance of the trend, relative variance of the seasonal, and first order autocorrelation coefficient of the residual. The remaining two aspects are an estimate of the spectral entropy, and the

estimated Box-Cox parameter. Ref. [36] refer to the entropy measure as "forecastability", and, rather unusually, autocorrelation in the residuals as "linearity" and Box-Cox as "stability".

These aspects are not so helpful to us as a forecaster: we wish to decide whether to use logarithms or not (i.e. an additive versus multiplicative model), whether there is seasonality or not, and if first differences are a transformation that helps with forecasting. Ideally we would also like to know if the series has been subject to large breaks, and perhaps the presence of nonlinearity.

### 3.2. Sample Size

Yearly, quarterly, and monthly data have a small number of series with a very large sample size, which is unlikely to benefit forecasting. Figure 1 shows the distribution of sample size at each frequency. Yearly, quarterly, monthly, and weekly data have been truncated to 40 years of data, and daily data to 1500 observations.

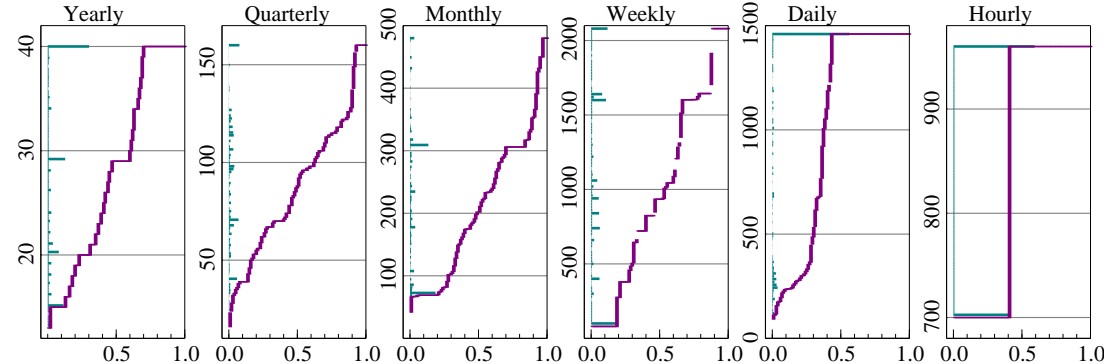

**Figure 1.** Distribution of sample size after truncation. Truncated at 40 years, except daily at 1500 observations.

Along the vertical axis is the bar chart of the frequencies of the different sample sizes. So for annual data about a third has sample size 40 after truncation, and the next biggest group has 29 observations. The same information is shown cumulatively as well in the line that runs from zero at the bottom left, to unity in the top right. This dual perspective reveals the central tendency without hiding the tails.

The truncation has a relatively large impact on daily data. Hourly data only comes in two sample sizes.

### 3.3. Logarithms

The benchmark methods do not take any transformations of the variables. In contrast, ref. [37] use a Box–Cox transformation (see [38]) in their bagging method. Ref. [39] improve on the *Theta* method in the M4 competition by using a Box–Cox transformation. In both cases the transformation is restricted to $\lambda \in [0,1]$, where $\lambda = 0$ corresponds to the logarithmic transformation and otherwise:

$$y_t(\lambda) = \lambda^{-1}\left(y_t^\lambda - 1\right).$$

So $\lambda = 1$ indicates the absence of any transformation. The difference between their two approaches is that the former does the Box–Cox transformation before deseasonalization, and the latter afterwards. We found that this distinction matters little: the values of $\lambda$ estimated before or after multiplicative seasonal adjustments are similar: in quarterly M4 and monthly M3 the correlation between the estimates exceeds 0.9.

Figure 2 shows the histogram of $\lambda$ estimated by maximum likelihood in a model on a constant and trend (and restricted to be between zero and one), using samples truncated as described in the previous section. This amounts to minimizing the adjusted variance as a function of $\lambda$. The U shapes in Figure 2 indicate that the choice is mainly between

logarithms and levels (Ref. [35] find a similar U shape for M3). This suggests a simpler approach, such as comparing the variance when using levels to that using logs. Removing the trend by differencing leads to an approach as in [40], i.e., using logs when $\min y_t > 1$ in combination with:

$$\exp(2\overline{\log y})\mathrm{var}[\Delta \log y_t] < c_l^2 \mathrm{var}[\Delta y_t], \tag{7}$$

where $\mathrm{var}[x_t]$ is the sample variance of $x_t, t = 1, \ldots, T$ and $\overline{\log y}$ is the sample mean of $\log y_t$. Using (7) means that the iterative estimation of $\lambda$ can be avoided.

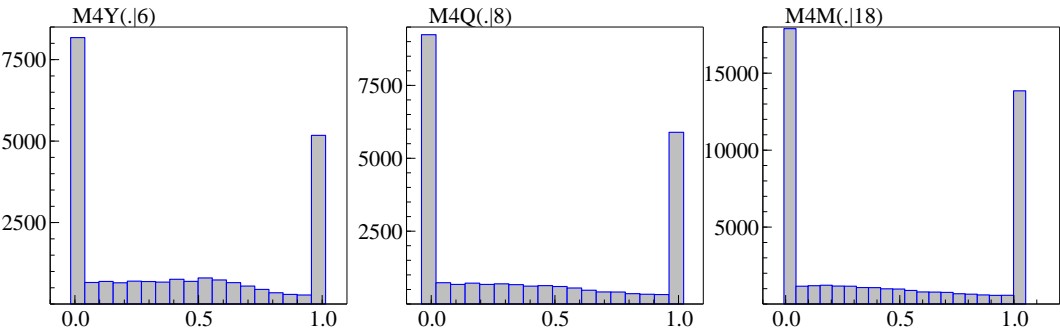

**Figure 2.** Estimated Box–Cox $\lambda$ for yearly (**left**), quarterly (**middle**) and monthly M4 (**right**)

All observations in M3 and M4 are positive, and experimentation suggests a benefit from preferring logs when $\widehat{\lambda} < 1$. The $c_l^2 > 1$ term in (7) is introduced to allow a bias towards using logarithms.

*3.4. Persistence*

To assess persistence, we estimate $\widehat{\rho}$ by OLS from

$$\log y_t = \mu + \rho \log y_{t-1} + \text{seasonal dummies} + \epsilon_t.$$

The seasonal dummies are excluded if the sample size is small ($T < 3S$) and for daily data.

The distribution of the autoregressive coefficient is plotted in Figure 3 for each frequency. The labels along each vertical axis correspond to the value of $\widehat{\rho}$, while the frequencies and proportions are on the horizontal axes.

The estimates of $\rho$ cluster near unity at all frequencies, corresponding to very high persistence in the data. However, more than 10% is above unity for annual data, which is likely caused by small samples. To a lesser extent this is found at the higher frequencies as well. Almost all daily series have a unit root.

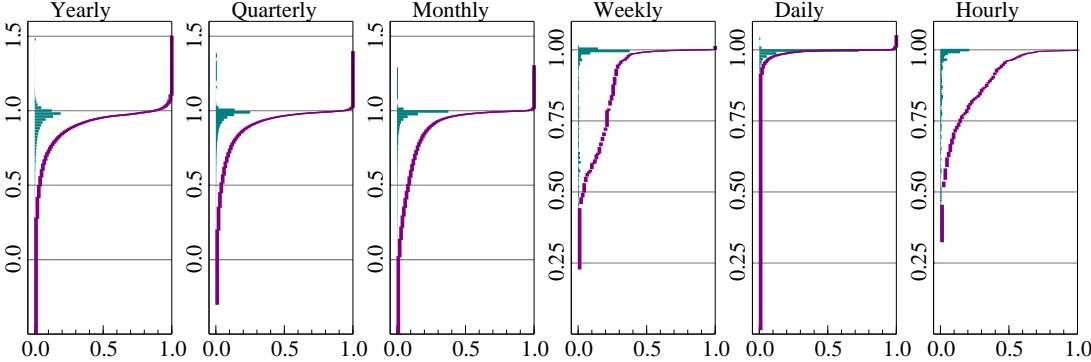

**Figure 3.** Distribution of autoregressive coefficient for each frequency of M4.

Except for hourly data, the seasonal adjustment does not have much visual impact on the graphs.

*3.5. Seasonality*

The top row of Figure 4 shows the distribution of the *p*-values of the test for a seasonal root (2), applied to the original $y_t$ (without sample truncation). This is the method used in the benchmarks, with the exception of daily data that we gave $S = 5$ to highlight the impact on inference. The vertical axis has the *p*-values, and testing quarterly data at 10% would say that almost 90% is seasonal. For monthly data it is close to 70% and for daily data almost all is classified as seasonal at a frequency of five.

The null hypothesis for test (2) assumes that there is no significant lower order serial correlation, which is mostly proven wrong by Figure 3. As a consequence, the incidence of seasonality is over-estimated, most strongly for daily data. A more accurate representation is to apply the test to $\Delta \log y_t$. This is shown in the middle row of graphs, now with a lower incidence of seasonality where almost none of the daily data, again tested with $S = 5$, appear to have seasonality.

X-11, see [41], incorporates an *analysis of variance*-based (ANOVA) test for stable seasonality, which has as the null hypothesis that the $S$ seasonal means are equal:

$$A\{y_t, S\} = \frac{S-1}{S(Y-1)} \frac{\text{var}[\bar{y}_{.,j}]}{\text{var}[y_{i,j}] - \text{var}[\bar{y}_{.,j}]} \sim \mathsf{F}[S-1, S(Y-1)]. \tag{8}$$

The sample size is adjusted to have $Y$ years with $S$ periods, and $\bar{y}_{.,j}$ denotes the mean of the observations pertaining to season $j$. The bottom row of graphs shows the *p*-values of the ANOVA test for stable seasonality, applied to $\Delta \log y_t$ and as used in [12].

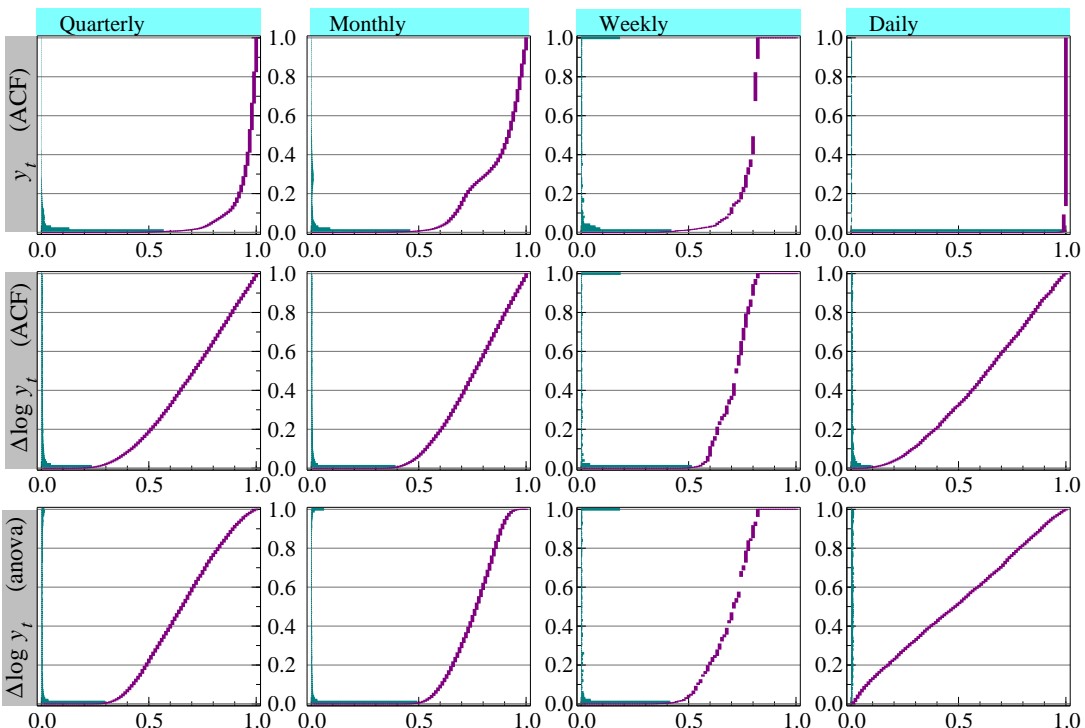

**Figure 4.** Tests of seasonality for quarterly, monthly, weekly and daily (with $S = 5$) M4. First row for $y_t$, second row for $\Delta \log y_t$, third row seasonal ANOVA test for $\Delta \log y_t$.

## 4. Revisiting the M4 Benchmark Methods

Section 4.1 approximates the expected values of the accuracy measures for three simple data generation processes (DGP) using random walk forecasts. Further details are

in Appendix A. Section 4.2 proposes a simplified version of *Theta(2)*, further improved in M3 forecasting by using logarithms.

### 4.1. Expected Performance of Naive Forecasts

Disregarding seasonality, the naive, or random walk, forecasts extrapolate the last-observed value into the future. For this device we can study the expected values of the two performance measures sMAPE (3) and MASE (4).

The MASE is invariant to a change in location and scale of the target variable, while the sMAPE is invariant to rescaling $y_t$ but not to a change in mean. They are sensitive to other transformations, say differences versus levels ([20]), and need not give the same ranking of forecast performance.

The sMAPE was introduced to address two issues with the MAPE (the MAPE is (3) but with just $|y_t|$ in the denominator): instability when outcomes close to zero are possible, as well as asymmetric response when exchanging outcomes and forecasts. However, sMAPE introduces a new problem, favoring overforecasting. As an illustration, take $y_t = \mu$, $\mu > \delta > 0$ with forecast $\mu + \delta$ or $\mu - \delta$:

$$\text{sMAPE}(\mu + \delta) = \frac{2\delta}{2\mu + \delta} < \text{sMAPE}(\mu - \delta) = \frac{2\delta}{2\mu - \delta}.$$

This was already noted by [42,43]. A second problem, in common with the MAPE, is that it has a bias that can be very large in some settings, illustrated in Appendix A.

For a better understanding, we derive approximate expectations of MASE and sMAPE when using random walk (naive) one-step forecasts. Three data generation processes (DGP) are considered: normally distributed in levels, stationary in differences and stationary growth rates. The results are derived in the Appendix A, and summarized in Table 2. In the third case, the MASE is very roughly proportional to the sample size.

**Table 2.** Approximate expectations of sMAPE and MASE under different data generation processes, $H = 1$. $\Phi$ is the standard normal cdf, $\phi$ the density, $m_3 = \exp(\mu + \sigma^2/2)$.

| DGP | sMAPE | MASE |
|---|---|---|
| $y_t \sim \text{IN}[\mu, \sigma^2]$ | $113\frac{\sigma}{|\mu|}$ | $1$ |
| $\Delta y_t \sim \text{IN}[\mu, \sigma^2]$ | $\frac{200}{2T+1}\left[2\frac{\sigma}{\mu}\phi\left(\frac{-\mu}{\sigma}\right) + 1 - 2\Phi\left(\frac{-\mu}{\sigma}\right)\right]$ | $1$ |
| $\Delta \log y_t \sim \text{IN}[\mu, \sigma^2]$ | $200\frac{m_3-1}{m_3+1}$ | $\frac{m_3^T}{\frac{1}{T}\sum_{t=1}^{T} m_3^{t-1}}$ |

The sMAPE behaves very differently. In the white noise case it is a fixed multiple of the coefficient of variation (inverse signal-to-noise ratio). In the second case, difference stationary, the sMAPE is inversely proportional to the sample size. Finally, in the third case, it is largely independent of sample size again. Table 3 gives the average MASE and sMAPE of the random walk forecasts of the last observation of the training sample for annual M3 and M4.

**Table 3.** Average MASE and sMAPE of 1-step Naive forecasts, forecasting the last observation of the training sample.

| | Yearly M3 (H = 1) | | Yearly M4 (H = 1) | |
|---|---|---|---|---|
| | sMAPE | MASE | sMAPE | MASE |
| *Naive2* | 9.585 | 1.416 | 8.390 | 1.688 |

### 4.2. A Simplified Theta Method: THIMA and THIMA.log

It could be convenient to have a simplified version of *Theta*(2) in our forecasting toolbox. Remember that SES is an ARIMA(0,1,1) model, and that slope of the trend

can also be estimated through the mean of the first differences. This leads us to suggest a trend-halved integrated moving average model (*THIMA*):

(1) Starting from $y_t$, $t = 1, \ldots, T$, the first differences $\Delta y_t$, $t = 2, \ldots, T$ have mean $\widetilde{\tau}$. Construct $x_t = \Delta y_t - \frac{1}{2}\widetilde{\tau}$.

(2) Estimate the following MA(1) model by nonlinear least squares (NLS) with $\widehat{\theta} \in [-0.95, 0.95]$:

$$x_t = \epsilon_t + \theta\epsilon_{t-1}.$$

(3) The forecasts are:

$$\widehat{y}_{T+H} = y_T + \tfrac{1}{2}\widetilde{\tau}H + \widehat{\theta}\widehat{\epsilon}_T. \tag{9}$$

The forecasts from the MA(1) component are $\widehat{\theta}\widehat{\epsilon}_T$ for $H = 1$ and zero thereafter: their cumulation is constant (this is also the case for the SES forecasts in *Theta*(2)). So *THIMA* forecasts consist of a dampened trend (arbitrarily halved), together with an intercept correction estimated by the moving average model (see e.g., [44]). That estimation of one parameter does not have to be very precise, and the overall procedure is very fast.

Keeping seasonality the same as in the other benchmarks, but adding a decision on logarithms, the full *THIMA.log* method is:

1. if (2) signals seasonality at 10% continue with the deseasonalized series as in Section 2.3;
2. if (7) using $c_l = 1.3$ suggests logarithms, take logs;
3. forecast using *THIMA*;
4. exponentiate the forecasts if logs were used, then add seasonality if the variable was deseasonalized.

A preference towards logarithms is introduced by setting $c_l = 1.3$. As a consequence, the proportion that is not logs in M4 is about 3% for annual data, 2% for quarterly and monthly, and less than 1% for the remainder. When logs are taken $\widetilde{\tau}$ is the mean growth rate.

We also consider *Theta.log*, which is like *THIMA.log*, but with the forecast step replaced by the Theta(2) method, see Section 2.2.

Table 4 reports the M3 performance of the modified and new benchmark methods, showing that both are improvements over the standard *Theta(2)* method. Their relative ranking is somewhat unclear because dropping one observation at the end shows *THIMA.log* as the best performer.

**Table 4.** M3 performance of MASE and sMAPE for *Theta(2)* and revised benchmark methods. Lowest in **bold**.

| M3 | Yearly (H = 6) | | Quarterly (H = 8) | | Monthly (H = 18) | |
|---|---|---|---|---|---|---|
| | sMAPE | MASE | sMAPE | MASE | sMAPE | MASE |
| Full sample, holdback $H$ | | | | | | |
| *Naive2* | 17.88 | 3.17 | 10.03 | 1.25 | 16.77 | 1.04 |
| *Theta(2)* | 16.72 | 2.77 | 9.24 | 1.12 | 13.91 | 0.87 |
| *Theta.log* | **16.00** | **2.68** | **9.15** | **1.11** | **13.57** | **0.85** |
| *THIMA.log* | 16.10 | **2.68** | 9.19 | **1.11** | 13.75 | 0.86 |
| With last observation removed, holdback $H$ | | | | | | |
| *Naive2* | 18.57 | 3.31 | 9.54 | 1.22 | 16.11 | 1.01 |
| *Theta(2)* | 17.07 | 2.87 | 9.26 | 1.13 | 13.61 | 0.84 |
| *Theta.log* | 15.91 | 2.64 | 9.26 | 1.13 | **13.22** | **0.82** |
| *THIMA.log* | **15.61** | **2.57** | **9.07** | **1.10** | **13.22** | **0.82** |

## 5. Heterogeneity and Independence

Table 4 shows that dropping one observation from M3 switched the best method from *Theta.log* to *THIMA.log*, uniformly for both measures at all three frequencies. Given the much larger size of M4, our assumption was that this cannot happen there, but Section 5.1

shows otherwise. Section 5.3 attributes this heterogeneity to clustering in the sample end-points. Section 5.2 introduces a DGP that in many respects is similar to M4.

### 5.1. Unexpected Heterogeneity

As a first investigation we assess the accuracy of 1-step ahead forecasts using an expanding data sample. Starting with the full data set (including the evaluation sample initially held back), we first withhold 12 observations, forecasting one step ahead. With $T_i$ observations in the training set for series $i$, the full series has $T_i^F = T_i + H$ observations. The first 1-step forecast is for $T_i^F - 12$. Then the next is for $T_i^F - 11$, etc. In sequence we give these twelve one-step forecasts the $x$-coordinates $-12, \ldots, -1$.

Table 5 illustrates how the data are used. The first line applied when we were developing our M4 submission: we held back an additional $H$ observations to test performance. The next line relates to the evaluation of the competition. The final three are for the sequence of 1-step forecasts that were just described. In each case the test samples (denoted by non-italic T) are the forecasts evaluated in the competition.

**Table 5.** Data use in developing, evaluating and forecasting for our M4 submission.

| | $1 \cdots\cdots T_i - H$ | $T_i\ -H\ +1 \cdots\cdots T_i$ | $T_i + 1 \cdots\cdots T_i + H$ |
|---|---|---|---|
| development | training | Test forecasts | unavailable |
| competition | competitor forecasts from this | | M4 team tests |
| first 1-step forecast | estimation | T       unused | |
| second 1-step forecast | estimation |      T      unused | |
| last 1-step forecast | estimation | | T |

Figure 5 reports the average sMAPE and MASE for the sequence of 1-step forecasts for M3 and M4 for four forecast devices. The lines are normalized by the *Naive2* forecast measures, so *Naive2* becomes a line at unity.

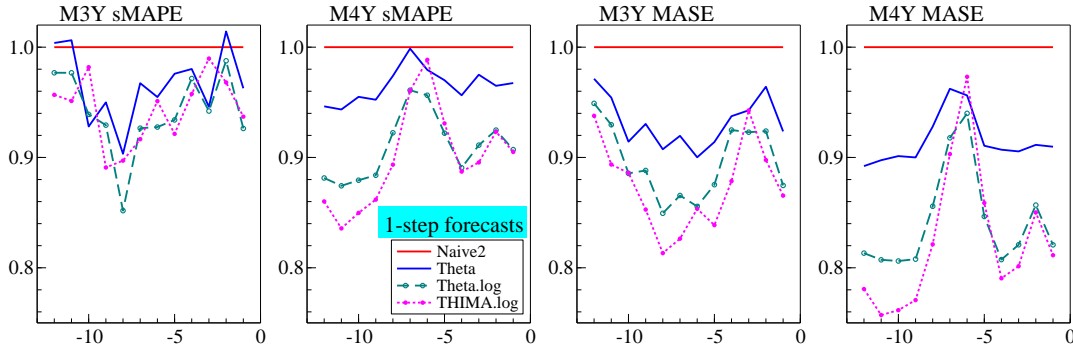

**Figure 5.** Average one-step ahead forecast accuracy for yearly M3 and M4, withholding from 12 to 1 observations at the end from the full datasets. Normalized by the naive results.

First we note from the Figure 5 graphs that the variability is similar between M3 and M4, despite moving from 645 to 23,000 series for annual data. This is unexpected, since the averages are computed over many more series. So uncertainty in rankings in M4 could be similar to that in M3. Next, accuracy rankings can switch for different subsamples, so it may not be enough to establish success by looking at one particular sample. This is not captured by the statistical comparisons of [5,23] because that is for a fixed sample, assuming the series are independent. Finally, the M3 profile is somewhat U shaped, but M4 is more the opposite. This is relevant, because the competition version omits the last six observations, which corresponds to the middle of the graphs.

Figure 5 also shows that all methods in the graph are, with a few exceptions, an improvement over the random walk forecast. As expected *THIMA.log* and *Theta.log* are similar, and both are largely improvements over the standard *Theta(2)* method. Their relative ranking is unclear in the graphs, as it was in Table 4.

### 5.2. An M4-Like Data Generation Process

As a first step in designing a DGP that mimics some of the properties of M4, we address normality. The backbone of our *Card* method ([12]), is the calibration step. We apply this calibration to the yearly and quarterly series without forecasting, and collect the residuals, standardized by their estimated equation standard error. This gives 630,515 yearly and 2,010,696 quarterly residuals. Figure 6 shows that normality is strongly rejected: with so many residuals, the 95% error bands (see [45]) are very tight. Normality is matched well in the center between $\pm 2$, but the residuals have fatter tails. This is not a surprise for economic data, where breaks happen intermittently. It also corresponds to the need to inflate the forecast intervals from calibration.

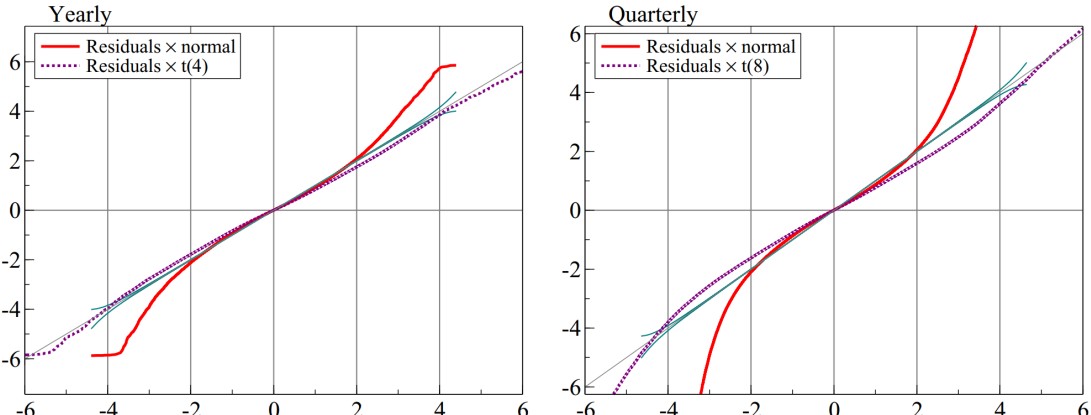

**Figure 6.** QQ plots of annual and quarterly residuals against Normal and closely matching Student-t distribution, t(4) for yearly and t(8) for quarterly data.

The proposed data generation process is:

$$x_t = \mu_0 U_1 + \rho x_{t-1} + \sigma[\delta_t + \epsilon_t + \theta \epsilon_{t-1}], \quad \epsilon_t \sim \text{IN}[0,1], \quad t = -99, \dots, T$$
$$U_1, U_2 \sim \text{IN}[1,1],$$
$$\sigma = \sigma_0 + 0.02(U_2^2 - 1),$$
$$\delta_t = 2u_t I(|u_t| > 2.58) \quad u_t \sim \text{IN}[0,1], t \geq 1 (\delta_t = 0 \text{ for } t < 1),$$
$$y_t = 100 \exp(x_t - x_1).$$

(10)

The autoregressive parameter is $\rho$, and the moving average parameter is $\theta$, while $\delta$ adds intermittent breaks. The DGP has a warm-up of 100 periods, during which there can be no breaks. The breaks occur in 1% of observations.

With parameters $\rho = 1, \theta = 0, \mu_0 = 0.03, \sigma_0 = 0.06$, this closely matches the M4 yearly data in terms of the mean and standard deviation of growth rates, the distribution of the estimated autoregressive coefficient, as well as the shape of the QQ plot of the calibration residuals. The dominance of the unit root ($\rho = 1$) in logarithmic form was already established from Figure 3, the values of MASE and sMAPE, as well as the improved benchmark methods.

For a superficial comparison, we show in Figure 7 eighteen real M4 series, followed by the same number of simulated series. They look comparable, except that the third series would be unlikely to arise from the DGP. The M4 data set has a few extremely large breaks that will not be replicated. Finally, the DGP could be lacking some heteroscedasticity and nonlinearity. Seasonality is still to be added.

However, the DGP (10) also has some advantages over M4. The first is that the data series have independent errors, but being drawn from the same DGP, are nevertheless highly correlated. This means that 'whole database' forecast methods do not inadvertently use future information, thus avoiding infeasible forecasts. Furthermore, generating data this way is much easier to implement, so can serve as an initial forecasting testbed. The

DGP captures the properties that seem relevant for economic time series, which can help to improve machine learning methods in such a setting. Finally, there are some parameters that can be varied, in addition to the sample size.

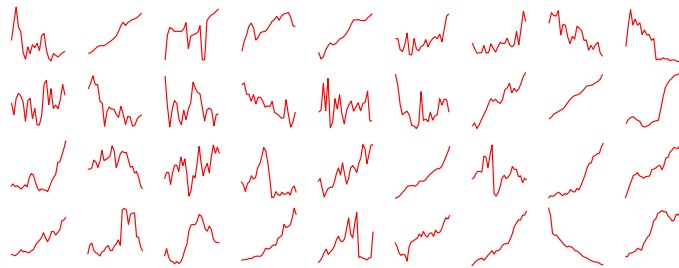

**Figure 7.** Eighteen actual yearly M4 series and eighteen simulated series.

Figure 8 compares the 1-step forecasts using the simulated data and annual M4. The simulated data has $T_i^F = 32$ observations for all series, and we use 10,000 series. The simulated data shows less variability in the MASE and sMAPE. The fact that the MASE is trending for the simulated data corresponds with the results in Section 4.1.

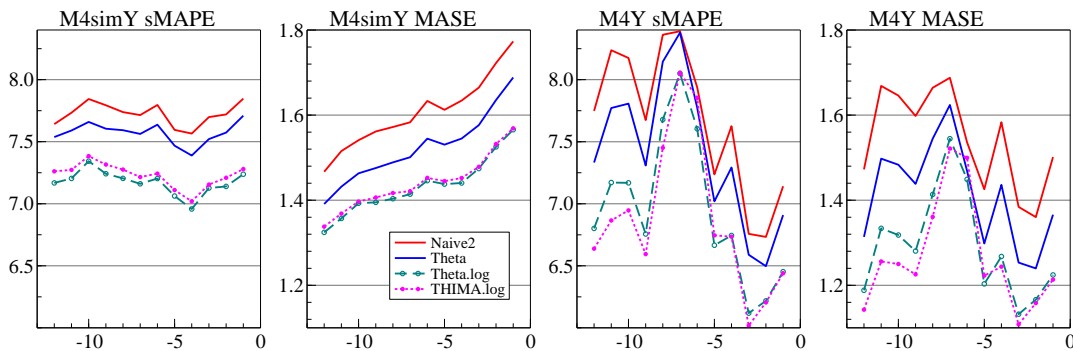

**Figure 8.** Average one-step ahead forecast accuracy for DGP and M4, withholding from 12 to 1 observations at the end from the full datasets.

### 5.3. The Role of Sample Dates

After the competition, the sample dates for the training series were supplied. Table 6 lists the sample ends for the annual M4 data where they comprise 1% or more. The ten listed dates together make up 94% of the annual sample. The year 2009 is the most common at over a quarter of the sample. Next is 2008, and together they are almost half of the sample. So most forecasts were for 2010–2015 and 2009–2014. These are the years in and after the financial crisis, which may contribute to the unexpected heterogeneity in the data.

**Table 6.** Fraction of the M4 annual data with the specified year end.

| End year | 1991 | 2001 | 2003 | 2004 | 2005 | 2006 | 2007 | 2008 | 2009 | 2010 |
|---|---|---|---|---|---|---|---|---|---|---|
| Part of sample | 15.2% | 1.6% | 3.9% | 4.3% | 1.1% | 1.1% | 5.2% | 21.7% | 26.2% | 14.0% |

## 6. The Cardt Method

Our submission to the M4 competition was labeled *Card*, summarized in Section 6.1. We introduce some small adjustments to that method, first a robust adjustment in Section 6.2, then adding *THIMA* to create *Cardt* in Section 6.3. We also derive interval forecasts that can be used at different significance levels and sample sizes in Section 6.4.

### 6.1. The Original Card Method

*Card* combined two autoregressive methods, incorporating trend dampening, some robustness against structural breaks, and limiting the autoregressive coefficient to prevent wild forecasts. The forecasts were obtained from a final 'calibration' step. [12] provide a technical description.

The first forecast method in *Card* is based on estimating the growth rates from first differences—hence labeled *Delta* method—but with removal of the largest values and additional dampening. The second method, called *Rho*, estimates a simple autoregressive model, possibly switching to a model in first differences with dampened mean.

Next, the forecasts of *Rho* and *Delta* are averaged with equal weights. Calibration treats the hold-back forecasts as if they were observed, and re-estimates a model that is a richer version of the first-stage autoregressive model (see (A5) in Appendix B). The fitted values over the forecast period (now pseudo in-sample) are the new forecasts. At this stage there is no issue with overfitting or explosive roots, because no further extrapolation is made.

Calibration makes little difference for annual or daily data, which have no seasonality. It does, however, provide almost uniform improvements in all other cases, in some cases substantially so. This experience is also reported for the X12-ARIMA procedure ([21]), although there the ARIMA model comes first, providing a forecast extension that is used in the X11 procedure. Our procedure could be a flexible alternative.

*Card* made initial decisions about the use of logarithms, differences versus levels, and the presence of seasonality:

1. Let $y_t, t = 1, \ldots, T$ denote the initial series. If $\min(y_1, \ldots, y_T) > 1$: $x_t = \log(y_t)$, else $x_t = y_t$.
   This entails that logs were always used in both the M3 and M4 data.
2. If $\text{var}[\Delta x_t] \leq 1.2 \, \text{var}[x_t]$ then forecast from a dynamic model, else directly forecast the levels using a static model.
   The static model only occurs in M4 at a rate of 1.5% (yearly), 4% (quarterly), 6% (monthly), and almost never at the other data frequencies.
3. The presence of seasonality is tested at 10% based on the ANOVA test (8) using $\Delta x_t$ or $x_t$ (depending on the previous step).

In our application to M4, we limited the number of observations of annual, quarterly, and monthly data to 40 years, and daily data to 4 years. The choice of 40 was a compromise between recency to avoid distortion from distant shifts yet sufficient data for parameter estimation. The longer the estimation sample, the greater the possibility of distributional shifts in the earlier data that would be inimical to forecasting. For hourly data, the *Rho* and *Delta* are calibrated, then averaged, then calibrated again; calibration is done with autoregressive lag six instead of one. For weekly data, *Rho* is applied to the four-weekly averages (giving frequency 13), and calibrated before averaging with *Delta*.

For hourly data, $S = 24$, we add a second frequency $S_2$ to reflect the diurnal rhythm: $S_2 = 7$ creates an additional frequency of $SS_2 = 168$ for the weekly pattern. In [12] we specified the daily frequency as $S \times S_2 = 5 \times 12$, but this was not beneficial, and we change it here to $S = S_2 = 1$. So no distinction is made anymore between yearly and daily data.

The Ox ([46]) code to replicate our Card submission was uploaded to Github shortly after the M4 competition deadline.

### 6.2. Robust Adjustments to Card

#### 6.2.1. Robust 1-Step Forecasts of AR(1) Model

A correction can make the forecast more robust when there is an unmodeled break at the forecast origin. To illustrate, consider a stationary autoregressive model of order one, AR(1):

$$y_t = \mu + \rho y_{t-1} + z'_t \beta + \epsilon_t, \quad t = 1, \ldots, T.$$

In the current setting, all components in $z_t$ are deterministic, so known for the forecast period (see [17] for an analysis where the future $z_t$'s are not known). The one-step forecast is

$$\widehat{y}_{T+1} = \widehat{\mu} + \widehat{\rho} y_T + z'_{T+1} \widehat{\beta}.$$

The robust forecast is taken from the differenced model:

$$\widehat{y}^R_{T+1} = y_T + \widehat{\rho} \Delta y_T + \Delta z'_{T+1} \widehat{\beta} = \widehat{\mu} + \widehat{\rho} y_T + z'_{T+1} \widehat{\beta} + y_T - \widehat{\mu} - \widehat{\rho} y_{T-1} - z'_T \widehat{\beta} = \widehat{y}_{T+1} + \widehat{\epsilon}_T.$$

The robust forecast is an intercept correction based on the last residual. When nothing changes, $E[\widehat{y}_{T+1}] = E[\widehat{y}^R_{T+1}]$, but the variance is increased by $\widehat{\sigma}^2_\epsilon$. However, if there is a location shift in $\mu$ at $T$, this is captured in the residual, so there is a trade-off between the increased variance and the reduction in bias. A seasonal equivalent can be based on the seasonally differenced model:

$$\widehat{y}^{R(S)}_{T+1} = \widehat{y}_{T+1} + \widehat{\epsilon}_{T+1-S}. \tag{11}$$

6.2.2. One-Step Ahead Robust Adjustment for *Rho*

Let $[x]^b_a$ denote the value of $x$ truncated to be between $a$ and $b$. The following adjustment is to the 1-step forecasts, but only if *Rho* is not already using first differences:

$$R = \left[ \tfrac{1}{2} \left( \widehat{\epsilon}_T + \widehat{\epsilon}_{T-S+1} \right) \right]^{+2\widehat{\sigma}}_{-2\widehat{\sigma}},$$
$$\widehat{x}^R_{T+1} = \widehat{x}_{T+1} + \tfrac{1}{2} R. \tag{12}$$

$R$ is the averaged residual following (11), which is limited to two residual standard errors. Half of that is added to the one-step forecast.

Experimentation with M4 shows that the robust version of *Rho* using (12) shows no benefit for monthly and weekly data: here the gains and losses are similar. The benefit is substantial for quarterly, daily, and hourly data. It is small but consistent for annual data. The results for M3 are similar: no improvement for monthly data, although the annual improvement is more pronounced.

6.2.3. One and Two-Step Robust Adjustment after Calibration

At very short horizons, calibration performs worse than the inputs to calibration. We therefore made a small change for the first two forecasts, taking the average of the original and calibrated forecasts. Beyond $H = 3$, the forecasts are unchanged at the fitted values from calibration.

*6.3. Cardt: More Averaging by Adding THIMA*

Our annual forecasts using *Card* did not do as well as expected. Part of the explanation is that holding back twelve observations from the full M4 data set is quite different from withholding six, as was illustrated in Figure 5. At the time we considered adding a Theta-like forecast to the average prior to calibration, but decided against this. That was a mistake, and *Cardt* for frequencies up to twelve now uses the calibrated average of *Rho*, *Delta*, and *THIMA.log*.

*6.4. Forecast Intervals*

The forecast intervals of an AR(1) model with drift, $y_t = \mu + \rho y_{t-1} + \epsilon_t$, grow with the horizon $h$ when the errors are IID:

$$SE = \sigma \left( 1 + \rho^2 + \ldots + \rho^{2(h-1)} \right)^{1/2}.$$

This is slower than the mean effect:

$$y_{T+h} = \mu\left(1 + \rho + \ldots + \rho^{h-1}\right) + \rho^h y_T.$$

Our submitted approach was based on $x_t = \log y_t$ with 90% forecast interval:

$$\exp\left[\widehat{x}_{T+h} \pm C_1 \widehat{\sigma}\left(1 + \widehat{\rho}_L + \ldots + \widehat{\rho}_L^{h-1}\right)\right],$$

where $\widehat{\rho}_L$ is an adjusted estimate of the autoregressive parameter and $C_1$ is determined by withholding data from the competition data set, aiming for a 90% interval. Even though this approach did well, it suffers from being asymptotically invalid, as well as being fixed at the 90% interval.

Our new approach makes a small-sample adjustment to the standard formula for the forecast interval. The basis for this is the calibration formula (see (A5) in Appendix B), which, however, is somewhat simplified and restrained. Because the sample size is small in some cases, we account for parameter uncertainty. Ideally, we get the correct point-wise coverage at each interval and in total, as well as a small MSIS from (6).

The forecast bands have the following form:

$$\left[\widehat{L}_{T+h}, \widehat{U}_{T+h}\right] = \left[\exp\left(\widehat{x}_{T+h} \pm c_\alpha\left\{(\text{var}[\widehat{x}_{T+h}])^{1/2} + \frac{\pi_h(S)}{T}\right\}\right)\right]. \tag{13}$$

This is the standard forecast uncertainty for an autoregressive model with regressors, but here with an inflation factor $\pi_h$ that depends on the frequency. The value of $\pi_h$ is determined by withholding $H$ observations from the training data, and finding a value that combined reasonably good coverage at all forecast horizons with a low value of MSIS. The form of $\pi_h$ is given in Appendix B. The critical value $c_\alpha$ is from a Student-t distribution, with the degrees of freedom given below (A7) in Appendix B.

In addition, for $S = 4, 12, 52$, we average the forecast standard errors from calibration in logs with those from calibration applied to the levels.

## 7. Evaluation

Section 7.1 considers the contribution of averaging and calibration on *Card*. Note that *Card* includes the robust adjustments of Section 6.2, unlike the version used in our M4 submission. The difference between these two versions of *Card* is small.

### 7.1. Averaging and Calibration

We start by considering the impact of averaging and calibration on the updated *Card* method. As in Figure 5, the sample is expanded one observation at a time, but this time the focus is on measuring joint performance over forecasts $1, \ldots, H$ (see Table 1 for the value of $H$). The holdback runs from $2H$ to $H$ observations, which is $12, \ldots, 6$ for annual data. This is labeled $-12, \ldots, -6$ on the horizontal axis of the graphs. The full data sets are used, so the forecasts with $H$ observations held back (the right-most point in each plot) would correspond to the forecasts evaluated in the competition.

Figure 9 shows the performance in terms of sMAPE for the data that are not seasonal. From left to right, these are 10,000 simulated series from DGP (10) (using same data as in Figure 8), annual M4, annual M3, and daily M4. In these cases, the MASE and sMAPE have a very similar profile. The graphs are standardized by *Naive2*. First we see that *Delta* is better than *Rho*, except for M3, where there is no difference. Otherwise, there is little difference between the average *DelRho* and the calibrated average *Card*. The main difference between the simulated data and M4 is the relative non-constancy of the latter. For daily data (labeled M4D), it is difficult to beat the random walk, as expected from data that is largely financial.

Figure 10 has the results for seasonal M4 data. Now averaging is mostly an improvement, and calibration improves further. The calibrated average seems a particularly

effective way to handle the complex seasonal patterns of weekly and hourly data. The seasonality in the MASE for hourly data is caused by remaining seasonality in the denominator of (4).

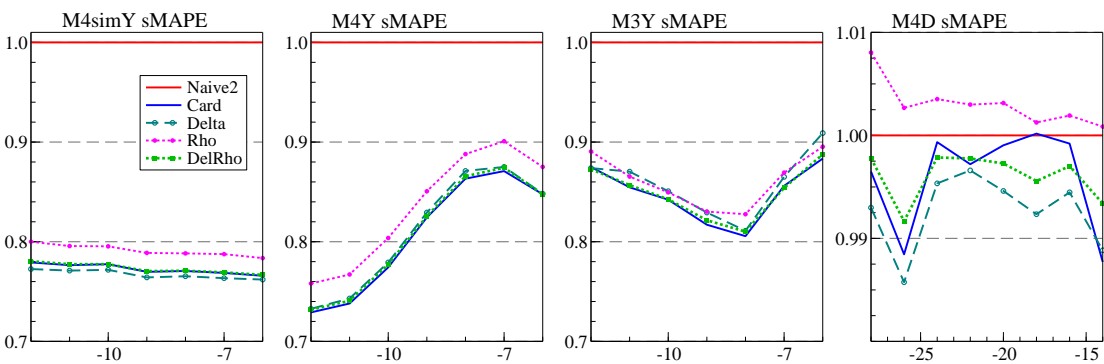

**Figure 9.** *H*-step sMAPE relative to that of *Naive2* for all non-seasonal data (simulated, M4, M3, daily M4), retaining from 2*H* to *H* observations for evaluation. Forecast methods are *Delta*, *Rho*, (*Delta* + *Rho*)/2, *Card*.

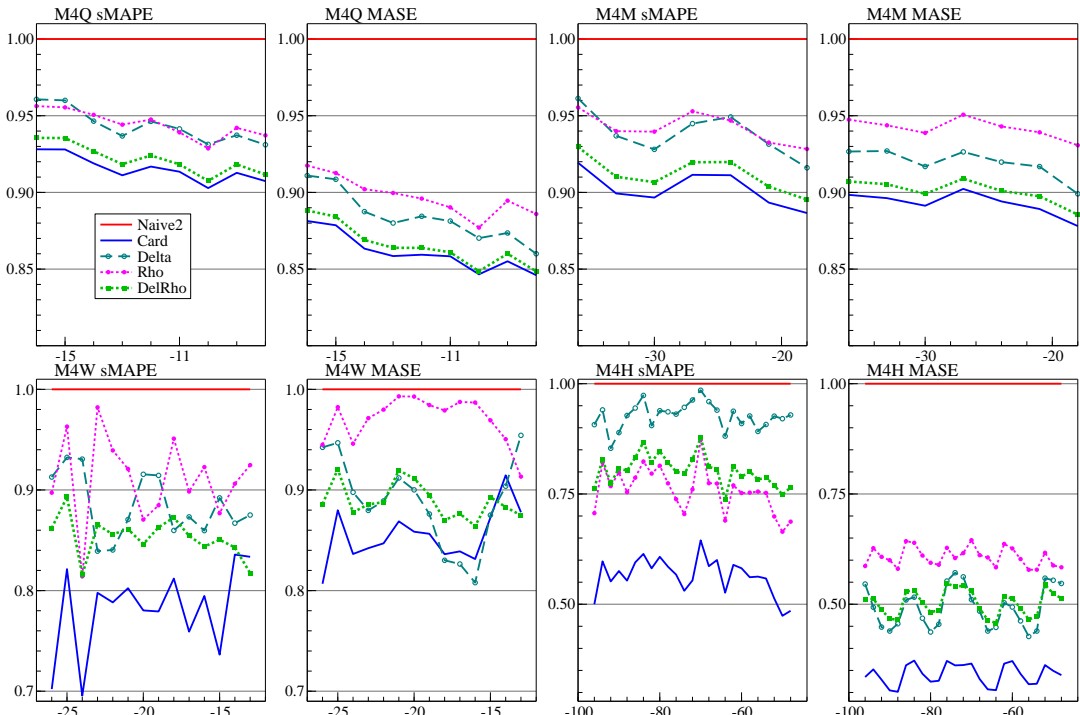

**Figure 10.** *H*-step performance relative to that of *Naive2* for all seasonal data (M4Q, M4M, M4W, M4H for quarterly, monthly, weekly, hourly), retaining from 2*H* to *H* observations for evaluation. Forecast methods are *Delta*, *Rho*, (*Delta* + *Rho*)/2, *Card*.

### 7.2. Comparison of Cardt and Card

Figure 11 looks at *THIMA.log*, which is added to *Card* at monthly or lower frequencies to give *Cardt*. This shows that the addition of *THIMA.log* makes *Cardt* consistently outperform *Card*. The top row of Figure 11 is for M4, and the bottom for M3. *THIMA.log* on its own is mostly worse, although there are some instances where it has the lowest sMAPE.

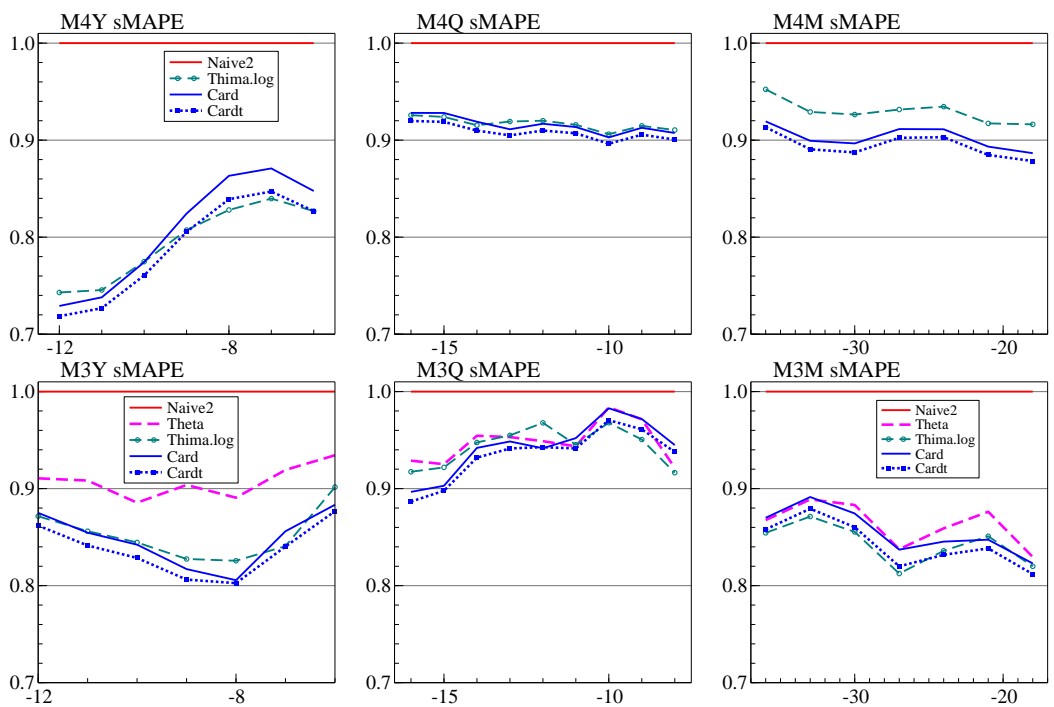

**Figure 11.** *H*-step forecast accuracy relative to that of *Naive2*. Forecast methods are *Card*, *Cardt*, *THIMA.log*.

### 7.3. Interval Forecasts

Figure 12 shows the coverage of the forecast intervals, averaged up to *H*-steps ahead for $\alpha = 0.05$ and $\alpha = 0.1$. The forecast intervals are generally well behaved. The exceptions are that the 90% intervals for monthly data are a bit too wide, and hourly intervals a bit too narrow. The bands' effectiveness fluctuates with the subsample, perhaps more than expected.

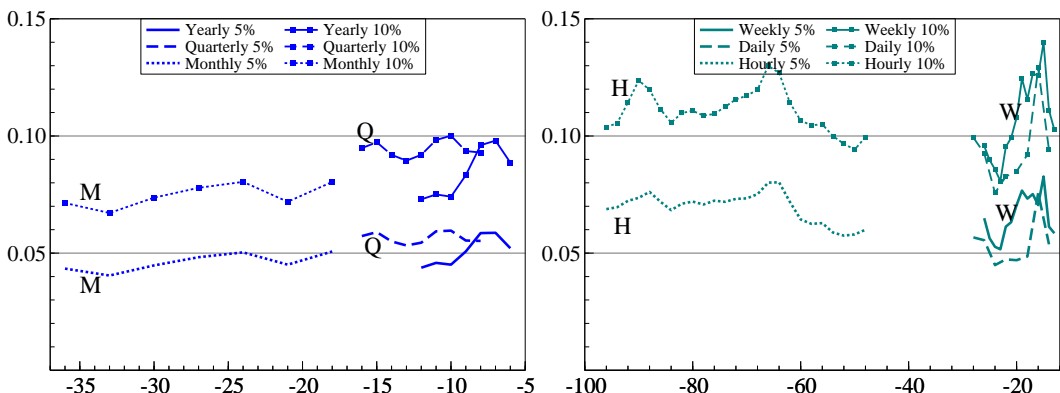

**Figure 12.** Average rejection of 95% and 90% *H*-step forecast intervals for all frequencies of M4, retaining from 2*H* to *H* observations for evaluation.

### 7.4. Overall Performance

Table 7 lists the forecast performance in terms of MASE and sMAPE for each frequency, as well as the weighted average for all frequencies combined. The overall weighted average (OWA), computed as in (5), is in the final column. The new *Cardt*, which adds *THIMA.log* to the forecast combination prior to calibration, is mainly improved for annual data, and just a bit better for quarterly data.

Interval forecasts are evaluated through the MSIS scores (6), which are reported in the bottom half of Table 7. The new method of computing the intervals is generally better. Also reported is the absolute coverage difference (ACD), which is the absolute difference between the fraction of outcomes outside the interval forecasts and the target significance level. The optimal value for the ACD is zero. Our M4 submission targeted 10% intervals, so we computed the ACD relative to that significance level for submitted *Card*. The overall ACD takes the weighted average, and then the absolute value, so is not equal to the weighted average of the ACDs at each frequency. The new approach to computing forecast intervals is comparable in terms of coverage, but a considerable improvement as measured by MSIS. The new intervals have been derived after the competition finished—nonetheless, they improve on our old intervals, which were already among the best.

**Table 7.** Summary performance in M4 competition. Absolute coverage difference (ACD) is for a 95% forecast interval except for submitted *Card* which used 90%. OWA is the overall weighted average of sMAPE and MASE, with weights determined by the relative number of series for each frequency. * denotes used ACD at 90%.

| M4 | Y | Q | M | W | D | H | Y | Q | M | W | D | H | All | | |
|---|---|---|---|---|---|---|---|---|---|---|---|---|---|---|---|
| | | | sMAPE | | | | | | MASE | | | | sMAPE | MASE | OWA |
| new *Cardt* | 13.51 | 9.91 | 12.67 | 6.75 | 3.01 | 8.92 | 3.10 | 1.15 | 0.93 | 2.33 | 3.21 | 0.81 | 11.757 | 1.582 | 0.849 |
| submitted *Card* | 13.91 | 10.00 | 12.78 | 6.73 | 3.05 | 8.91 | 3.26 | 1.16 | 0.93 | 2.30 | 3.28 | 0.80 | 11.924 | 1.627 | 0.865 |
| new *THIMA.log* | 13.51 | 10.02 | 13.21 | 7.90 | 3.03 | 18.41 | 3.05 | 1.17 | 0.97 | 2.54 | 3.24 | 2.50 | 12.090 | 1.601 | 0.864 |
| | | | MSIS | | | | | | ACD 90%*/95% | | | | MSIS | ACD | |
| new *Cardt* | 25.72 | 8.92 | 8.23 | 16.01 | 27.01 | 5.84 | 0.002 | 0.000 | 0.003 | 0.007 | 0.005 | 0.013 | 13.23 | 0.002 | |
| submitted *Card* * | 30.20 | 9.85 | 9.49 | 16.47 | 29.13 | 6.14 | 0.013 | 0.021 | 0.004 | 0.003 | 0.009 | 0.048 | 15.18 | 0.007 | |

## 8. Cardt and COVID-19

*Cardt* is a relatively simple forecasting device, based on some of the principles discussed here: robustness, trend dampening, averaging, limited number of parameters with limited range in the first stage, followed by a richer model in the final calibration. *Cardt* is fast and performs well on both M3 and M4 data, suggesting that it can be a general purpose device to provide baseline forecasts for macro-economic data. Ref. [17] shows in simulations that *Cardt* can be useful to forecast conditioning variables in the presence of breaks. Ref. [47] apply this to unemployment forecasting, showing that the conditional model has more accurate forecasts for 2019, but *Cardt* for 2020.

One unresolved issue is that of the forecast horizon. Because of the calibration, and as an example, the third forecast for a horizon of three is not identical to the third for a horizon of four. Next is the use of logarithms. In M3 and M4 it was best (and easiest) to always use logs. But a default of testing using (7) with $c_l = 1.3$ may be preferable in other settings.

We performed some additional robustness checks on the performance of Cardt. The first set of experiments vary the parameters and sample size of the artificial DGP. The second is for a range of transformations of the M4 variables, including logs, differences, differences of logs, square roots, reversing each series, and cumulating each series. Cardt outperforms *Naive2*, *Theta2*, *THIMA.log*, *Delta*, and *Rho* in almost all cases. The one exception is for the cumulated M4 series, where *Rho* tends to perform badly. This will be caused by the suppression of the autoregressive parameter and the trend. In turn this affects *Cardt*, especially for monthly, weekly, daily and hourly data. The *Delta* method does surprisingly well for the cumulated data, suggesting a route for improvement.

While the cumulated M4 series do not look much like economic data, they have more in common with the cumulated counts of cases and deaths from COVID-19: this pandemic data shows periods of rapid growth followed by slowdowns, subjected to many shocks, including shifts in measurements and definitions with large ex post revisions. As documented in [48], we use a novel trend-seasonal decomposition of cumulated counts. Forecasts of the estimated trend and remainder are then made with the *Cardt* method. For

the trend we can apply *Cardt* to the differenced trend, and then cumulate the forecasts, say *Cardt*-I(2). At the start of the pandemic, although data was still limited, this was found to overestimate the future growth. As a solution we adopted the average of *Cardt* and *Cardt*-I(2). Refs. [48,49] investigate the forecast performance of the adjusted *Cardt* method for COVID-19, showing generally good performance.

## 9. Conclusions

We established that the dominant features of M4 are mostly stationary growth rates that are subject to intermittent large shocks, combined with strong seasonality. This led us to propose a simple extension to the *Theta* method by adding a simple rule to take logarithms. We also introduced *THIMA.log* as a simple benchmark method that helps understanding the *Theta* method. Moreover, this improves on *Theta(2)* in both M3 and M4 forecasting at low frequencies. We added this to the forecast combination prior to calibration, which mainly improved performance at the yearly frequency.

Our experience with M4 supports most of the principles that were introduced in the introduction:

(I)     dampen trends/growth rates;
        This certainly holds for our methods and *Theta*-like methods. Both *Delta* and *Rho* explicitly squash the growth rates. *Theta(2)* halves the trend. The *THIMA* method that we introduced halves the mean of the differences, which has the same effect.

(II)    average across forecasts from "non-poisonous" methods;
        This principle, which goes back to [16], is strongly supported by our results, as well as the successful methods in M4. There may be some scope for clever weighting schemes for the combination, as used in some M4 submissions that did well. It may be that a judicious few would be better than using very many.
        A small amount of averaging also helped with forecast intervals, although the intervals from annual data in levels turned out to be 'poisonous.'

(III)   include forecasts from robust devices in that average;
        We showed that short-horizon forecasts of *Rho* could be improved by overdifferencing when using levels. The differenced method already has some robustness, because it reintegrates from the last observation. This, in turn, could be an adjustment that is somewhat too large. The IMA model of the *THIMA* method effectively estimates an intercept correction, so has this robustness property (as does *Theta(2)*, which estimates it by exponential smoothing).

(IV)    select variables in forecasting models at a loose significance;
        Some experimentation showed that the seasonality decisions work best at 10%, in line with this principle. Subsequent pruning of seasonal dummies in the calibration model does not seem to do much, probably because we already conditioned on the presence of seasonality. However, for forecast uncertainty, a stricter selection helps to avoid underestimating the residual variance. Ref. [17] find support for this in a theoretical analysis.

(V)     update estimates as data arrive, especially after forecast failure;
        This aspect was only covered here by restricting estimation samples to say, 40 years for annual data given the many large shifts that occurred in earlier data. Recursive and moving windows forecasts are quite widely used in practical forecasting.

(VI)    'shrink' estimates of autoregressive parameters in small samples;
        As the forecast error variance can only be estimated from out-of-sample extrapolation, it is essential to avoid explosive behaviour, so constrain all $\hat{\rho} \leq 1$.

(VII)    adapt choice of predictors to data frequency;
        For example, method 118 by [50] had the best performance for yearly and monthly forecasting but *Card* was best at forecasting the hourly data.

(VIII)   address 'special features' like seasonality.

Appropriate handling of seasonality was important as described in Section 2.3 and even transpired to be an important feature of forecasting COVID-19 cases and deaths as in [49].

We derived a DGP that generates data similar to annual M4. This could be a useful complement to the actual data. It also confirmed the good performance of our *Cardt* method. Extending this to include the properties of the seasonal time series is left to a later date.

We believe that *Cardt* is useful as a baseline method for economic forecasting: it is fast, transparent, and performs well in a range of realistic and simulated settings. Most recently, it helped to provide useful short-run forecasts of COVID-19 cases and deaths.

**Author Contributions:** Conceptualization, J.L.C., J.A.D. and D.F.H.; Data curation, J.L.C. and D.F.H.; Formal analysis, J.L.C., J.A.D. and D.F.H.; Funding acquisition, D.F.H.; Investigation, J.L.C. and D.F.H.; Methodology, J.L.C., J.A.D. and D.F.H.; Resources, D.F.H.; Software, J.A.D.; Validation, D.F.H.; Writing—original draft, J.L.C. and D.F.H.; Writing—review and editing, J.L.C. and D.F.H. All authors have read and agreed to the published version of the manuscript.

**Funding:** Financial support from the Robertson Foundation (award 9907422), Nuffield College and Institute for New Economic Thinking (grant 20029822) and the ERC (grant 694262, DisCont) is gratefully acknowledged.

**Institutional Review Board Statement:** Not applicable.

**Informed Consent Statement:** Not applicable.

**Data Availability Statement:** https://github.com/Mcompetitions/M4-methods https://forecasters. org/resources/time-series-data/m3-competition/ (accessed on 25 January 2021).

**Acknowledgments:** We thank two anonymous referees for their helpful suggestions.

**Conflicts of Interest:** The authors declare no conflict of interest.

## Appendix A. Accuracy Measures under Naive Forecasts

The random walk (or naive) forecast of annual data is simply $\widehat{y}_{T+1} = y_T$, so for one step ahead MASE, from (4):

$$\text{MASE(N1)} = \frac{|\Delta y_{T+1}|}{\frac{1}{T}\sum_{t=1}^{T}|\Delta y_t|}. \tag{A1}$$

Similarly for MAPE and sMAPE:

$$\text{MAPE(N1)} = 100\frac{|\Delta y_{T+1}|}{|y_{T+1}|}, \tag{A2}$$

$$\text{sMAPE(N1)} = 200\frac{|\Delta y_{T+1}|}{|y_{T+1}| + |y_T|}. \tag{A3}$$

In this setting we can derive approximate expected values of these measures.

*Appendix A.1. Stationary Case*

First assume that the data generation process is given by:

$$y_t = \mu + \epsilon_t, \quad \epsilon_t \sim \text{N}[0, \sigma^2].$$

Then $\Delta y_t \sim \text{N}[0, 2\sigma^2]$, therefore $|\Delta y_t|$ has a half normal distribution, and

$$\text{E}[|\Delta y_t|] = \sigma\left(\frac{4}{\pi}\right)^{1/2} = 2\sigma\phi(0) \equiv m_1.$$

Using the first term of the Taylor expansion around the expectation amounts to approximating the expectation of the ratio by the ratio of the expectations:

$$E[\text{MASE(N1)}] \approx \frac{E[|\Delta y_{T+1}|]}{\frac{1}{T}\sum_{t=1}^{T} E[|\Delta y_t|]} \approx 1.$$

The numerator and denominator of the MASE(N1) in (A1) are asymptotically uncorrelated because the distributions of $|\Delta y_t|$ and $|\Delta y_s|$ are independent for $s < t - 1$.

For the denominator of sMAPE, note that $|y_t|$ has a folded normal distribution:

$$E[|y_t|] = 2\sigma\phi\left(\frac{-\mu}{\sigma}\right) + \mu\left[1 - 2\Phi\left(\frac{-\mu}{\sigma}\right)\right] \equiv m_2.$$

So

$$E[\text{sMAPE(N1)}] \approx 100\frac{m_1}{m_2}.$$

When $\mu/\sigma$ is large enough:

$$E[\text{sMAPE(N1)}] \approx 100\frac{m_1}{\mu} = 113\frac{\sigma}{|\mu|}.$$

This does not hold when $\mu = 0$: in that case the expectation is approximately $100\sqrt{2} = 141$. The numerator and denominator of the sMAPE (A3) are uncorrelated provided $y_{T+1}$ and $y_T$ have the same sign (an alternative version with $|y_{T+1} + y_T|$ in the denominator would always be uncorrelated).

The approximations can provide some insight in the amount of bias introduced when minimizing the error measures: what is the optimal $\beta$ to add to the naive forecast in the current DGP? In the MASE, only the nominator is affected, turning it into a folded normal distribution. Then minimizing the approximation amounts to minimizing $f(\beta) = 2\phi(-\beta) + \beta[1 - 2\Phi(-\beta)]$. Because $\partial f(\beta)/\partial\beta = 1 - 2\Phi(-\beta)$, this is zero at $\beta = 0$: the bias comes from the higher order terms that were ignored in the approximation.

In case of the sMAPE, the bias function is more difficult because $\beta$ also enters the denominator. For $\mu > 0$, sMAPE is roughly minimized for $\beta = \sigma^2/\mu$. Table A1 compares the approximations to simulations for a range of means when the variance equals one, confirming the increasing accuracy of the approximate expectations as $\mu$ increases.

**Table A1.** Simulated and approximated mean and bias of MASE and sMAPE for one-step ahead naive forecasts. DGP $N[\mu, 1]$, $T = 15$, M = 100,000 replications.

| | $\mu = 0.5$ | $\mu = 1$ | $\mu = 2$ | $\mu = 5$ | $\mu = 10$ |
|---|---|---|---|---|---|
| | | | Simulated | | |
| E[MASE] | 1.136 | 1.136 | 1.136 | 1.136 | 1.136 |
| E[sMAPE] | 133.9 | 109.1 | 63.2 | 23.0 | 11.3 |
| Bias[MASE] | −0.006 | −0.006 | -0.006 | −0.006 | −0.006 |
| Bias[sMAPE] | 1.100 | 0.923 | 0.573 | 0.198 | 0.093 |
| Bias[MAPE] | −0.279 | −0.932 | −1.640 | −0.222 | −0.108 |
| Bias[MAAPE] | 0.064 | 0.038 | −0.109 | −0.159 | −0.100 |
| | | | Approximated | | |
| E[MASE] | 1 | 1 | 1 | 1 | 1 |
| E[sMAPE] | 225.7 | 112.8 | 56.4 | 22.6 | 11.3 |
| Bias[MASE] | 0 | 0 | 0 | 0 | 0 |
| Bias[sMAPE] | 2 | 1 | 0.5 | 0.2 | 0.1 |

Table A1 also shows that while the MASE is essentially unbiased, the bias from minimizing sMAPE and MAPE is large in some cases. The MAAPE, introduced by [51], is defined as:

$$\text{MAAPE} = \frac{100}{H} \sum_{t=T+1}^{T+H} \text{atan2}(|y_t - \hat{y}_t|, |y_t|) \tag{A4}$$

*Appendix A.2. Nonstationary Case*

Keeping $T$ fixed:

$$\Delta y_t = \mu + \epsilon_t, \quad \epsilon_t \sim \text{N}[0, \sigma^2], t = 1, \dots, T+1.$$

Then $|\Delta y_t|$ has a folded normal distribution with mean $m_2$ and $\text{E}[\text{MASE}] \approx 1$. Setting $y_0 = 0$ we have that $y_t = \sum_{s=1}^t \Delta y_s \sim \text{N}[t\mu, t\sigma^2]$.

$$\text{E}[\text{sMAPE(N1)}] \approx 200 \frac{m_2}{(2T+1)\mu}.$$

This is roughly $100/T$ when $\mu/\sigma$ is large, but more like $(100/T)\sigma/\mu$ for small $\mu$.

*Appendix A.3. Nonstationary Case in Levels*

Let

$$y_t = \exp(x_t),$$
$$\Delta x_t = \mu + \epsilon_t, \quad \epsilon_t \sim \text{N}[0, \sigma^2],$$
$$x_t = \sum_{s=1}^t \Delta x_t, \quad x_0 = 0.$$

For fixed $T$, $\exp(\Delta x_t)$ has a lognormal distribution with mean $\exp(\mu + \sigma^2/2) \equiv m_3$, and $y_t = \exp(x_t)$ has a lognormal distribution with mean $\exp(t\mu + t\sigma^2/2) = m_3^t$, so

$$\text{E}[\Delta y_t] = \text{E}[[\exp(x_t)] - \text{E}[[\exp(x_{t-1})]] = m_3^{t-1}(m_3 - 1).$$

Ignoring the absolute values:

$$\text{E}[\text{MASE}] \approx \frac{m_3^T}{\frac{1}{T} \sum_{t=1}^T m_3^{t-1}}.$$

This is always positive, because $m_3 - 1$ cancelled out in this approximation. As a consequence, the approximation remains somewhat effective even for negative $\mu$. Note that the MASE tends to zero as $\mu$ gets more negative. For larger $\mu$ the MASE behaves as $Tm_3$.

Finally, for the sMAPE when $\mu$ is large:

$$\text{E}[\text{sMAPE(N1)}] \approx 200 \frac{m_3^{T+1} - m_3^T}{m_3^{T+1} + m_3^T} = 200 \frac{m_3 - 1}{m_3 + 1}.$$

**Table A2.** Simulated means and standard deviations of MASE, sMAPE, MAPE, and MAAPE for one-step ahead naive forecasts. $T = 15$, $M = 100,000$ replications.

| | $y_t \sim N[\mu, \sigma^2]$ | | $\Delta y_t \sim N[\mu, \sigma^2]$ | | $\Delta \log y_t \sim N[\mu, \sigma^2]$ | |
|---|---|---|---|---|---|---|
| | **Mean** | **Sdev** | **Mean** | **Sdev** | **Mean** | **Sdev** |
| | $\mu = 0, \sigma = 1, T = 15$ | | | | | |
| MASE | 1.14 | 0.92 | 1.04 | 0.84 | 2.9 | 9.8 |
| sMAPE | 144.1 | 68.5 | 51.0 | 58.9 | 69.9 | 45.4 |
| MAPE | 6629.4 | $18 \times 10^5$ | 809.6 | $1.6 \times 10^5$ | 112.4 | 195.1 |
| MAAPE | 90.8 | 40.1 | 40.3 | 38.8 | 58.7 | 37.9 |
| | $\mu = 0.025, \sigma = 0.1, T = 15$ | | | | | |
| MASE | 1.14 | 0.92 | 1.05 | 0.85 | 1.4 | 1.2 |
| sMAPE | 141.3 | 69.3 | 37.7 | 50.0 | 8.3 | 6.3 |
| MAPE | 654.0 | 10871 | 365.7 | 15478 | 8.5 | 6.8 |
| MAAPE | 89.5 | 40.6 | 31.7 | 34.8 | 8.4 | 6.7 |
| | $\mu = 0.1, \sigma = 1, T=15$ | | | | | |
| MASE | 1.14 | 0.92 | 1.04 | 0.84 | 4.3 | 13.2 |
| sMAPE | 143.8 | 68.6 | 48.5 | 57.6 | 70.1 | 45.5 |
| MAPE | 797.4 | 23462 | 188.7 | 6304.6 | 124.6 | 219.1 |
| MAAPE | 90.6 | 40.2 | 38.6 | 38.2 | 60.8 | 39.6 |
| | $\mu = 1, \sigma = 1, T = 15$ | | | | | |
| MASE | 1.14 | 0.92 | 1.04 | 0.75 | 41.8 | 71.6 |
| sMAPE | 109.1 | 71.0 | 8.60 | 7.02 | 94.0 | 51.6 |
| MAPE | 648.1 | 27743 | 9.31 | 21.5 | 358.8 | 579.4 |
| MAAPE | 75.3 | 43.5 | 9.07 | 7.52 | 91.9 | 46.8 |
| | $\mu = 10, \sigma = 1, T = 15$ | | | | | |
| MASE | 1.14 | 0.92 | 1.00 | 0.104 | $5.1 \times 10^5$ | $6.6 \times 10^5$ |
| sMAPE | 11.3 | 8.63 | 6.90 | 0.69 | 200.0 | 0.039 |
| MAPE | 11.5 | 9.05 | 7.15 | 0.74 | $36 \times 10^5$ | $47 \times 10^5$ |
| MAAPE | 11.3 | 8.71 | 7.14 | 0.74 | 157.1 | 0.010 |

## Appendix B. Forecast Intervals

Forecast intervals are obtained from the calibration model that is used to create the final forecasts. The calibration model is:

$$
\begin{aligned}
z_t &= \mu + (\rho z_{t-1} + \rho_R z_{t-R} I_R I_4 I_S + \rho_{R+1} z_{t-R-1} I_R I_4 I_S) I_\rho + \{\delta_j q_{j,t}\} I_A I_S \\
&+ (\gamma_1 S_t + \gamma_1^* C_t)(1 - I_A) I_S + (\gamma_2 S_{2,t} + \gamma_2^* C_{2,t})(1 - I_3) I_{S2} \\
&+ \rho_{SS_2} z_{t-SS_2} I_3 I_{S2} + (\tau_1 d_t + \tau_2 t d_t I_5 I_\rho) I_6 + u_t, \quad t = T_0, \dots, T + H
\end{aligned} \tag{A5}
$$

where $I_\rho = 0$ for a static model, $I_S = S > 1$, $I_R = 1$ when $R > 1$, $I_4 = T > 4S$, $I_3 = T + H > 4SS_2$, $I_{S2} = S_2 > 1$, $I_6 = S \neq 24$ and $T > 3S$ and $T + H - k > 10$, $I_5 = S = 4, 12, 13$, $S_t = \sin[2\pi t/S]$, $C_t = \cos[2\pi t/S]$, $S_{2,t} = \sin[2\pi t/(SS_2)]$, $C_{2,t} = \cos[2\pi t/(SS_2)]$, $d_t = I(t < T - \min[2S, (T + H)/2])$. Note that no observations are lost when lag $SS_2$ is used, because the first $SS_2$ observations are duplicated at the start.

The preliminary forecasts are treated as if they are in sample observations, and then replaced by fitted values from calibration. However, the forecast error variance can only be estimated from out-of-sample extrapolation. This makes it essential to avoid explosive behaviour; we also wish to avoid underestimating the residual variance, so (A5) is adjusted as follows:

1.  remove the broken intercept and trend (if present, so setting $I_6 = 0$);
2.  remove deterministic variables that are insignificant at 2%; the intercept is kept;

3. remove $z_{t-R-1}$ if present;
4. add the absolute residuals from (A5) as a regressor;
5. estimate the reformulated calibration model;
6. if $\widehat{\rho} > 0.999$ then impose the unit root, and re-estimate;
7. if $\widehat{\rho} < 0$ then set $\rho = 0$, and re-estimate.

Let $\widehat{u}_t$ denote the reformulated calibration residuals, then the equation standard error is estimated from 'recent' residuals:

$$\widetilde{\sigma}_u^2 = \sum_{\max(T-T^*+1, T_0)}^{T} \frac{\widehat{u}_t^2}{\max[\min(T^*, T - T_0 + 1 - k^*), 2]}, \quad T^* = \max(SS_2, 80), \quad \text{(A6)}$$

where $k^*$ is the number of regressors in the reformulated calibration model. The variance (A6) is computed from 'recent' residuals to reflect neglected (conditional) heteroscedasticity. $T$ is the original sample size excluding the forecast period, so the residuals from the forecast period are excluded; $T_0$ equals 1 for a static model, 2 for a model with one lag, and $R + 2$ if the seasonal lag is included.

The parameters of the adjusted model are estimated using all observations, but the forecast variance is extrapolated using the standard autoregressive forecast formulae from $T + 1$ onwards. This can be represented as

$$\widetilde{\sigma}_u^2 \left[ \widehat{f}_{T+h}^u + \widehat{f}_{T+h}^x \right],$$

where $f^u$ is the contribution from the error term, and $f^x$ the contribution of parameter estimation, with the former dominating asymptotically.

Two small adjustments are made: we use the recent residual variance (A6) and limit the contribution from parameter uncertainty:

$$\text{var}[\widehat{z}_{T+h}] = \widetilde{\sigma}_u^2 \left[ \widehat{f}_{T+h}^u + \min(\widehat{f}_{T+h}^x, 4\widehat{f}_{T+h}^u) \right].$$

By default, the modeling is in logs, so that the $100(1 - \alpha)\%$ interval is given by:

$$\left[ \widehat{L}_{T+h}, \widehat{U}_{T+h} \right] = \left[ \exp\left( \widehat{z}_{T+h} \pm c_\alpha \left\{ (\text{var}[\widehat{z}_{T+h}])^{1/2} + \frac{\pi_h(S)}{T} \right\} \right) \right]. \quad \text{(A7)}$$

The critical value $c_\alpha$ is from a Student-t distribution with $T - T_0 - k^*$ degrees of freedom.

The $\pi_h$ term is an inflation factor for $S = 4, 12, 24$ that is added when using logarithms because otherwise the forecast intervals are too small, particularly at longer horizons:

$$\pi_h(S) \begin{cases} 0.25h & S = 1, \\ 0.1h & S = 4, \\ 0.4h & S = 12 \\ 0.4\lfloor h/6 \rfloor & S = 24 \\ 0.0 & S = 52 \end{cases}$$

One further step for forecast intervals from calibration in logarithms is to also calibrate the levels, and then average the two. In that case the forecast standard errors are multiplied by $1 + 4h/T$. This is used for $S = 4, 12, 52$. For yearly data, the levels forecast intervals are too far out to be useful in a combination.

## Appendix C. Comparison with R Code

Table A3 compares *Theta2* forecasts using the R code supplied with the M4 competition to our Ox implementation (using all data). The forecast summary statistics are almost identical except for weekly data, where we get a different result. The Ox version is

about 130 times faster. Of that advantage, a factor of three is obtained from the parallel implementation.

**Table A3.** Timings to compute *Theta2* forecasts, comparing R and Ox version.

| *Theta2* | Ox Implementation | | R Implementation | | | |
|---|---|---|---|---|---|---|
| | Time (s) Total | sMAPE | Data | Time (s) Forecast | Total | sMAPE |
| Yearly | 3.67 | 0.876 | 4.61 | 375 | 380 | 0.880 |
| Quarterly | 4.83 | 0.949 | 4.31 | 627 | 631 | 0.950 |
| Monthly | 10.49 | 1.017 | 4.22 | 1499 | 1503 | 1.016 |
| Weekly | 0.45 | 0.838 | 3.95 | 33 | 37 | 0.886 |
| Daily | 6.96 | 1.007 | 4.05 | 1019 | 1023 | 1.008 |
| Hourly | 0.27 | 0.991 | 3.89 | 28 | 32 | 0.991 |

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
