# Peer review of "Forecasting Principles from Experience with Forecasting Competitions"

_forecasting, doi:10.3390/forecast3010010_

Round 1

Reviewer 1 Report

This is a good and competent contribution in the field of forecasting. It is not the first study on the M4 competition, so the added value could be a bit better elaborated.

In particular, if a main empirical result is dominance of a method called "delta" over a method called "rho", both should be described clearly, and the current description on p.15 is not optimal.

Minor issues collected:

p.4,l.136: These benchmark methods

p.5,l.161: tests reject, they are not rejected

p.9: I was a little surprised that X-11 is still so much in focus, I thought it had been superseded by X-12 or even 13 some time ago

p.11,l.202: Table 4 shows (it still does so)

p.14,l.347: independent albeit highly correlated: unclear phrase, as usually independence implies uncorrelatedness

Reviewer 2 Report

Please see attached review.
